# JSCDS: A Core Data Selection Method with Jason-Shannon Divergence for Caries RGB Images-Efficient Learning

Peiliang Zhang*
zhangpl109@whut.edu.cn
School of Computer Science and
Artificial Intelligence
Wuhan University of Technology
Wuhan, China

Yujia Tong*
tyjjjj@whut.edu.cn
School of Computer Science and
Artificial Intelligence
Wuhan University of Technology
Wuhan, China

Chenghu Du
duch@whut.edu.cn
School of Computer Science and
Artificial Intelligence
Wuhan University of Technology
Wuhan, China

Chao Che†
chechao@gmail.com
Key Laboratory of Advanced Design
and Intelligent Computing (Dalian
University), Ministry of Education
School of Software Engineering
Dalian University
Dalian, China

Yongjun Zhu
zhu@yonsei.ac.kr
Department of Library and
Information Science
Yonsei University
Seoul, Korea

## ABSTRACT

Deep learning-based RGB caries detection improves the efficiency of caries identification and is crucial for preventing oral diseases. The performance of deep learning models depends on high-quality data and requires substantial training resources, making efficient deployment challenging. Core data selection, by eliminating low-quality and confusing data, aims to enhance training efficiency without significantly compromising model performance. However, distance-based data selection methods struggle to distinguish dependencies among high-dimensional caries data. To address this issue, we propose a **Core Data Selection Method with Jensen-Shannon Divergence (JSCDS)** for efficient caries image learning and caries classification. We describe the core data selection criterion as the distribution of samples in different classes. JSCDS calculates the cluster centers by sample embedding representation in the caries classification network and utilizes Jensen-Shannon Divergence to compute the mutual information between data samples and cluster centers, capturing nonlinear dependencies among high-dimensional data. The average mutual information is calculated to fit the above distribution, serving as the criterion for constructing the core set for model training. Extensive experiments on RGB caries datasets show that JSCDS outperforms other data selection methods in prediction performance and time consumption. Notably, JSCDS exceeds the performance of the full dataset model with only 50% of the core data, with its performance advantage becoming more pronounced in the 70% of core data.

## KEYWORDS

Caries Classification, Core Dataset, Jason-Shannon Divergence, Image Processing, Deep Learning

**ACM Reference Format:**
Peiliang Zhang, Yujia Tong, Chenghu Du, Chao Che, and Yongjun Zhu. 2018. JSCDS: A Core Data Selection Method with Jason-Shannon Divergence for Caries RGB Images-Efficient Learning. In *Proceedings of Make sure to enter the correct conference title from your rights confirmation emai (Conference acronym 'XX).* ACM, New York, NY, USA, 5 pages. https://doi.org/XXXXXXX.XXXXXXX

## 1 INTRODUCTION

Dental caries is one of the chronic diseases affecting the health of children and adolescents. If not treated promptly, it can lead to complications such as pulpitis and periapical periodontitis, adversely affecting the development and quality of life of adolescents [9]. According to incomplete statistics, approximately 50% of children aged 6-11 have caries in their primary teeth, and over 34% of adolescents aged 12-19 have caries in their permanent teeth globally [2]. This underscores the importance of early detection and treatment of dental caries in maintaining the oral health of children and adolescents. Currently, the most widely used methods for caries detection are visual inspection, probing, and X-ray examination. However, these methods depend heavily on the professional expertise of dentists and increase the economic burden on patients. Therefore, efficient, accurate, and affordable caries detection methods are urgently needed.

**Previous Work:** According to different detection methods, caries detection can be categorized into physical examination, auxiliary examination, and automated detection [4, 5, 11]. Physical examination refers to dentists visually inspecting the tooth surface for changes

*The authors contributed equally to this research.
†The corresponding author.

**Unpublished working draft. Not for distribution.**

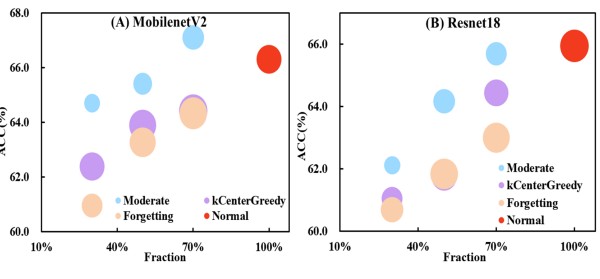

Figure 1: The motivation statement for JSCDS.

in color and shape to identify caries, with common methods including visual inspection and probing. While physical examinations are cost-effective, they rely heavily on the dentist's expertise, have a higher misdiagnosis rate, and require patients to visit the clinic, which is time-consuming and labor-intensive. Auxiliary examinations utilize modern medical equipment to scan or chemically test teeth for caries detection [11, 15]. Methods such as X-ray examination, laser fluorescence detection, electrical impedance measurement, and optical coherence tomography fall under this category. Although auxiliary examinations are more accurate compared to physical examinations, they are also significantly more expensive. Automated detection involves using automated tools to capture RGB images of teeth and intelligently determine the presence of caries [5]. With the advancement of Deep Learning, various caries recognition and detection models have been developed, making automated caries detection increasingly mainstream [6, 8]. Compared to the other two methods, deep learning-based automated caries detection offers significant advantages in terms of portability, high accuracy, and low cost.

**Limitation and Motivation:** Deep learning methods based on RGB images have significantly improved the efficiency and accuracy of caries detection while reducing its costs [15]. However, deep learning methods require high-quality and large training datasets. Low-quality, confusing data in the training dataset can severely affect the model's performance. In caries detection, patients primarily use mobile devices to capture RGB images of their teeth. During this process, due to the oral cavity's unique structure and the photographers' varying skill levels, the captured images often contain a certain amount of unclear, mixed-boundary confusing data. These data can severely impact the performance of the detection model [4, 5]. Data core set selection is one effective way to address this issue. By quantifying the impact of different data on the model, core data selection identifies key data and reduces the influence of confusing data, thereby enhancing the performance and efficiency of model training [7, 10, 16]. As shown in Figure 1(A), the Moderate method achieved superior performance and faster computational efficiency using 70% of the core dataset, with similar results observed in Figure 1(B). Therefore, how to measure the impact of different caries data on the detection model and selecting the core dataset accordingly is crucial for improving caries detection performance and computational efficiency.

To address the aforementioned problems, we propose a core data selection method with Jensen-Shannon Divergence (JSCDS) for efficient caries image learning and caries classification. Specifically,

JSCDS calculates the cluster centers of different classes with sample embeddings in the caries classification network. The core data selection criterion is described as the distribution of samples within different classes. We utilize Jensen-Shannon Divergence (JSD) to compute the mutual information between data samples and cluster centers and use average mutual information (AvgMI) as the data selection criterion. JSCDS selects samples close to the AvgMI as the core set for model training. Unlike distance-based data importance selection methods, JSCDS captures mutual information gain between samples, enhancing the handling of high-dimensional data. We conduct extensive experiments using MobileNetV2 and ResNet18, and the results demonstrate that JSCDS has lower time costs in data selection and exceeds the performance of the full dataset model using only 50% of the core data.

Our main contributions are as follows:

- Different from distance-based data selection methods, we focus on information theory-based data selection to identify high-dimensional nonlinear dependencies in samples. We propose the concept of AvgMI to generate the core set, effectively distinguishing between different caries.
- As a proof of concept, we design a mutual information calculation method with Jensen-Shannon Divergence. This method does not depend on model structure and selects core data only with the neural network's embedding.
- The results demonstrate that JSCDS significantly outperforms others in prediction performance and time consumption. Notably, JSCDS exceeds the performance of the full dataset model using only 50% of the core data.

## 2 METHODOLOGY

### 2.1 Definition

We define the core data selection problem in caries classification. For a given full dataset of caries training data $\mathcal{M} = \{m_1, m_2, \cdots, m_x\}$, which contains $x$ caries images. Any caries image $m_i \in (\mathcal{F}_i, \mathcal{T}_j)$, where $\mathcal{T}_j \in \{\mathcal{T}^{Mil}|\mathcal{T}^{Mod}|\mathcal{T}^{Sev}\}$ represents the image's label. We aim to design a data selection strategy $\mathcal{Y}(*)$ to construct a subset $\mathcal{M}^* = \{m_1^*, m_2^*, \cdots, m_y^*|y \leq x\}$ to remove confusing data, such as ambiguous images from the original dataset. Formally, $\mathcal{M} \xrightarrow{\mathcal{Y}(\mathcal{M})} \mathcal{M}^*$. When the model's performance on $\mathcal{M}^*$ is comparable to $\mathcal{M}$, and its training efficiency significantly improves, we refer to $\mathcal{M}^*$ as the core set and use it for model training.

### 2.2 Model Structure

To construct the core set for caries, we propose JSCDS with Jensen-Shannon Divergence. The workflow is shown in Figure 2. JSCDS primarily consists of two parts: data representation and data selection.

*2.2.1 Data Representation and Cluster Center Calculation.* For the neural network model for caries classification, it can be viewed as a black-box feature extractor. For the caries image $\mathcal{F}_i$, the probability distribution $\mathcal{P}(\mathcal{Z}, \mathcal{F}_i)$ of the image sample can be obtained through the mapping $\mathcal{Z}(*) = \mathcal{H}(h(*))$. Here, $h(*)$ represents the final embedding representation of the input caries image in the hidden layer, and $\mathcal{H}(*)$ is the feature mapping function represented by

**Figure 2: The workflow of JSCDS. JSCDS calculates cluster centers based on neural network embeddings representation, and then combines the AvgMI of data samples to generate the core set for model training.**

$Softmax(*)$. The hidden layer embedding representation of the training data $F_i$ can be obtained through the above calculation, and from this, the hidden embedding representation of all training data $\mathcal{M}$ can be derived, as shown in Equation (1). We calculate the cluster center $C_j$ for each data class based on the hidden layer representations by Equation (2).

$$\mathcal{M} = \{m_1, \cdots, m_x\} \xrightarrow{\mathcal{Z}(*)} \mathcal{E} = \{e_1 = h(\mathcal{F}_1), \cdots, e_x = h(\mathcal{F}_x)\} \quad (1)$$

$$C_j = \frac{\sum_{i=1}^{x} \mathcal{I}[T_i = j] \cdot e_i}{\sum_{i=1}^{x} \mathcal{I}[T_i = j]}, j = 1, 2, \cdots, J \quad (2)$$

Here, the numerator represents the sum of the sample embeddings in class $\mathcal{T}_j$, and the denominator represents the number of samples in class $\mathcal{T}_j$. The cluster center $C_j$ is the reference for subsequent data selection processes.

*2.2.2 Data Selection with JSD.* Relevant research primarily measures the relevance of samples by calculating the distance between them [10, 16]. While this method is computationally efficient, it heavily depends on the data size [7, 16]. Identifying high-dimensional nonlinear dependencies in the training data is challenging, making it difficult to distinguish between samples. In contrast, information theory-based methods can capture nonlinear relationships between samples, enabling effective high-dimensional data processing. Therefore, JSCDS designs a JSD-based data sample importance selection strategy from the information theory perspective.

Based on the hidden layer representations $\{e_1, e_2, \cdots, e_x\}$ of the training data and the cluster centers $\{C_1, C_2, \cdots, C_j\}$ of the classes, the mutual information $jsd(e, C)$ between each image embedding representation and its corresponding cluster center is calculated. $jsd(e, C)$ is obtained by computing the JSD:

$$jsd(e, C) = JSD(e \parallel C) \quad (JS \; divergence)$$
$$= \frac{1}{2} KL(e \parallel \mathcal{G}) + \frac{1}{2} KL(C \parallel \mathcal{G}) \quad (KL \; divergence)$$
$$= \frac{1}{2} \sum_w e(w) \log \frac{2 * e(w)}{e(w) + C(w)} \quad (\mathcal{G} = \frac{e+C}{2}) \quad (3)$$
$$+ \frac{1}{2} \sum_w C(w) \log \frac{2 * C(w)}{e(w) + C(w)}$$

where $e$ and $C$ represent the data samples and the class cluster centers, respectively. The JSD is computed by combining KL divergences, and $w$ denotes the embedding dimension of $e$.

The set of mutual information between each sample in the training set and its class cluster center is denoted as $MI(m_1), \cdots, MI(m_x)$, sorted in descending order to obtain $\{MI(\hat{m}_1), \cdots, MI(\hat{m}_x)\}$. We use the average mutual information $Avg(MI(m))$ as the metric, selecting samples close to $Avg(MI(m))$ as the core set $M^*$.

$$Avg(MI(m)) = (\sum_x MI(\hat{m}_x))/x \quad (4)$$

$$\{MI(\hat{m}_1) \; \cdots \; MI(\hat{m}_\gamma) \; \cdots \; Avg() \; \cdots \; MI(\hat{m}_{x-\gamma}) \; \cdots \; MI(\hat{m}_x)\}$$
$$\Updownarrow$$
$$\{\hat{m}_1 \quad \cdots \quad \hat{m}_\gamma \quad \cdots \quad \hat{m} \quad \cdots \quad \hat{m}_{x-\gamma} \quad \cdots \quad \hat{m}_x\}$$
$$(5)$$

Here, $\gamma$ represents the selection ratio of data in the core set (To ensure clarity in our descriptions, we describe $\gamma$ as "Fraction" in the experimental section.), and $\{\hat{m}_\gamma, \cdots, \hat{m}_{x-\gamma}\}$ is the core set after selection, which is used for model training.

In information theory, the more considerable mutual information between a data sample and the cluster center indicates the higher posterior probability given by the neural network for the sample. However, obtaining embedding representations in neural networks is easier than computing posterior probabilities [1]. Therefore, our method is more practical in real-world applications and requires fewer resources, thereby improving the computational efficiency of deep learning.

## 3 EXPERIMENTAL

### 3.1 Datasets

We used the RGB dental caries dataset labeled and processed by professional dentists. This dataset has been widely used in caries object detection and classification [4, 5]. It contains 5619 caries images, each in 24-bit JPG format. Professional dentists classified the caries images into three categories: mild, moderate, and severe caries, based on the extent and severity of the lesions. Following the settings in previous related studies [4, 5, 17], we divided the dataset into training, validation, and test set in the ratio of 8:1:1.

**Table 1: The caries classification results of JSCDS and comparison methods in different fractions.**

| Method | Fraction | MobilenetV2 [12] | | | | | | Resnet18 [3] | | | | | |
|---|---|---|---|---|---|---|---|---|---|---|---|---|---|
| | | ACC | Pre | Rec | F1 | SPE | Times | ACC | Pre | Rec | F1 | SPE | Times |
| − | 100% | 66.31 | 64.86 | 65.05 | 64.71 | 83.16 | 277 | 65.95 | 64.95 | 64.75 | 64.43 | 82.98 | 230 |
| random | 10% | 61.59 | 59.90 | 60.10 | 59.40 | 80.79 | 136 | 60.16 | 58.77 | 58.88 | 58.41 | 80.08 | 123 |
| Moderate [16] | | 62.03 | 63.62 | 61.88 | 62.03 | 81.02 | 144 | 62.12 | 62.14 | 61.51 | **61.66** | 81.06 | 130 |
| kCenterGreedy [13] | | 62.39 | 61.05 | 60.20 | 58.09 | 81.19 | 228 | 61.05 | 58.80 | 58.66 | 55.72 | 80.53 | 175 |
| Forgetting [14] | | 60.34 | 57.90 | 57.53 | 52.74 | 80.17 | 214 | 60.70 | 57.67 | 58.63 | 56.86 | 80.35 | 167 |
| JSCDS | | **64.62** | **64.00** | **63.82** | **63.85** | **82.31** | 145 | **64.35** | **62.41** | **62.56** | 61.42 | **82.17** | 146 |
| random | 30% | 62.03 | 60.29 | 60.54 | 59.88 | 81.02 | 169 | 62.92 | 60.94 | 61.49 | 60.97 | 81.46 | 150 |
| Moderate [16] | | 64.71 | 63.76 | 63.69 | 63.54 | 82.35 | 177 | 64.17 | 62.64 | 62.71 | 62.10 | 82.09 | 167 |
| kCenterGreedy [13] | | 62.39 | 59.73 | 60.49 | 59.08 | 81.19 | 274 | 61.76 | 59.91 | 59.25 | 56.07 | 80.88 | 203 |
| Forgetting [14] | | 60.96 | 55.90 | 57.93 | 52.03 | 80.48 | 230 | 61.85 | 59.14 | 59.08 | 54.62 | 80.93 | 192 |
| JSCDS | | **65.51** | **63.96** | **64.13** | **63.67** | **82.75** | 180 | **65.60** | **63.80** | **64.18** | **63.69** | **82.80** | 161 |
| random | 50% | 64.88 | 63.60 | 63.79 | 63.62 | 82.44 | 199 | 64.53 | 62.57 | 63.00 | 62.36 | 82.26 | 173 |
| Moderate [16] | | 65.42 | 64.49 | 64.47 | **64.47** | 82.71 | 209 | 65.69 | 63.90 | 64.02 | 63.15 | 82.84 | 183 |
| kCenterGreedy [13] | | 63.90 | 63.43 | 62.94 | 62.99 | 81.95 | 302 | 64.44 | 62.85 | 62.90 | 62.26 | 82.22 | 219 |
| Forgetting [14] | | 63.28 | 61.08 | 61.10 | 58.95 | 81.64 | 287 | 63.01 | 60.03 | 60.88 | 58.92 | 81.51 | 215 |
| JSCDS | | **66.49** | **64.99** | **65.00** | 64.45 | **83.24** | 208 | **66.13** | **65.05** | **65.16** | **65.05** | **83.07** | 182 |
| random | 70% | 66.58 | 66.36 | 65.97 | 66.08 | 83.29 | 231 | 66.13 | 64.77 | 64.95 | 64.71 | 83.07 | 190 |
| Moderate [16] | | 67.11 | 66.09 | 66.01 | 65.84 | 83.56 | 239 | 66.22 | 64.73 | 64.95 | 64.60 | 83.11 | 193 |
| kCenterGreedy [13] | | 64.44 | 62.67 | 63.04 | 62.54 | 82.22 | 312 | 65.06 | 63.33 | 63.54 | 62.84 | 82.53 | 247 |
| Forgetting [14] | | 64.44 | 62.09 | 62.58 | 61.34 | 82.22 | 308 | 64.62 | 62.58 | 62.90 | 61.95 | 82.31 | 239 |
| JSCDS | | **68.72** | **67.29** | **67.53** | **67.31** | **84.36** | 239 | **67.11** | **65.92** | **66.07** | **65.91** | **83.56** | 206 |

## 3.2 Experimental Setup

*3.2.1 Evaluation Metrics.* We constructed caries fine-grained classification to verify the data selection capability of JSCDS. In the caries classification task, Accuracy (ACC), Precision (Pre), Recall (Rec), F1-score (F1), Specificity (SPE), AUPR, and AUROC are commonly used to evaluate model performance [17–19]. AUPR and AUROC are closely related to evaluation metrics such as Pre and Rec. Due to page limitations, we use ACC, Pre, Rec, F1, and SPE to evaluate model performance in this work.

*3.2.2 Training Details.* We designed and implemented JSCDS based on Python and PyTorch, and conducted training and testing on a server equipped with two NVIDIA GeForce RTX 4090 GPUs (each with 24GB RAM), Ubuntu operating system and an Intel(R) Core(TM) i9-12900KF CPU. The backbone networks for the experiments were MobileNetV2 [12] and ResNet18 [3]. We initialize the model parameters using the model that has been pre-trained on the ImageNet-1K dataset. The training parameters for JSCDS are as follows: we train the network for 50 epochs using a learning rate of 0.001, no weight decay, batch size of 64, and Adam optimizer. Every 10 epochs,the core set is reselected.

## 3.3 Overall Performance

We compared the performance of JSCDS with four core set selection methods. The primary comparison methods include: random data selection, Moderate [16], kCenterGreedy [13], and Forgetting [14].

To comprehensively analyze the performance of JSCDS, we conducted extensive experiments on the dental caries classification dataset. The experimental results are shown in Table 1. The parameter settings of the comparison methods all followed the principle of optimal performance in our experimental equipment.

*3.3.1 Comprehensive Performance Analysis.* As shown in Table 1, our proposed JSCDS achieves the best fine-grained caries prediction results in different fractions. In predictive performance, JSCDS's performance on 50% of the core set is already close to or even exceeds the predictive results using the full dataset. At the 70% fraction, its predictive performance significantly surpasses that of the full dataset, as highlighted by the red data in Table 1. Figure 3 further visually confirms these findings. Regarding time overhead, JSCDS effectively reduces the training time, substantially enhancing training efficiency. In summary, JSCDS effectively selects high-quality data from the original dataset, improving the model's classification performance. The main reason is that JSCDS calculates the AvgMI with JSD and uses it to select the core set, capturing high-dimensional nonlinear dependencies among samples.

*3.3.2 The Performance Analysis of Different Core Set Selection Methods.* Compared to Moderate, kCenterGreedy, and Forgetting, JSCDS achieves the best comprehensive predictive performance in different backbone networks. Although the training time for JSCDS is marginally longer than that for Moderate, the substantial improvement in performance makes this additional time investment acceptable. Moderate's shorter training time can be attributed to its use of

Euclidean distance for data selection, which is computationally simpler. However, as the volume of selected data increases, Moderate's computational speed significantly decreases. This experimentation result aligns with the general knowledge that Euclidean distance incurs higher computational overhead in large-scale datasets. In contrast, JSCDS maintains its advantages in predictive performance and time overhead as the data scale grows. kCenterGreedy and Forgetting fail to achieve satisfactory predictive performance and training time, possibly due to the limited number of caries data classes. Moderate and JSCDS yield commendable results regarding training time overhead, significantly reducing the training time.

*3.3.3 The Performance Analysis in Different Fractions.* As shown in Table 1, all models achieve optimal predictive performance when the fraction is 50% or 70%, with minimal performance differences from the full dataset. JSCDS and Moderate even surpass the full dataset's performance at 70% fraction, indicating their effectiveness in core data selection. As the fraction increases from 30% to 70%, the training time for JSCDS only increases by 21% and 19.5% in MobilenetV2 and Resnet18, respectively. This demonstrates that JSCDS is efficient in data selection, with its time consumption not proportionally increasing with larger fractions. In contrast, kCenterGreedy and Forgetting have consistently high training times, and Moderate's performance in this aspect is slightly inferior to JSCDS. This result further underscores JSCDS's efficiency. The results in Figure 3 indicate that JSCDS achieves the best performance at a 70% fraction, with its predictive performance in both backbone networks showing a near-normal distribution trend. This suggests that the core set size significantly impacts model performance. The core set that is too large may include some noisy data, while a core set that is too small lacks sufficient high-quality data, both of which can degrade the model's predictive performance.

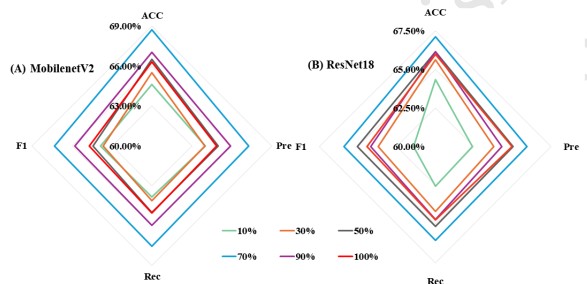

**Figure 3: The prediction results for different fractions. The red line indicates the results with the full dataset.**

## 4 CONCLUSION

In this paper, we designed a core set selection method with Jensen-Shannon Divergence. By capturing high-dimensional dependencies in caries images, we construct a high-quality core set for model training to improve the model's predictive performance. Extensive experiments on RGB caries datasets show that JSCDS outperforms other data selection methods in prediction performance and time consumption. Notably, JSCDS exceeds the performance of the full dataset model with only 50% of the core data, with its performance advantage becoming more pronounced in the 70% of core data.

## 5 ACKNOWLEDGMENTS

This work was supported by the National Natural Science Foundation of China (Grants No. 62076045), the Basic Project for Universities from the Educational Department of Liaoning Province (Grants No. LJKFZ20220290), the 111 Project (Grants No. D23006), the Fundamental Research Funds for the Central Universities (Grants No. 2023vb041) and the Interdisciplinary Project of Dalian University (Grants No. DLUXK-2023-YB-003 and DLUXK-2023-YB-009).

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
