# OpenReview forum: "JSCDS: A Core Data Selection Method with Jason-Shannon Divergence for Caries RGB Images-Efficient Learning"
_KDD.org/2024/Workshop/AIDSH — KDD-AIDSH 2024 Poster_

### Official Review · Reviewer_3Q6W · 2024-06-18
**Reviews from 3Q6W**

**Rating:** 6
**Confidence:** 4

**Review:**

This paper introduces a Core Data Selection Method based on Jensen-Shannon Divergence (JSCDS) to enhance the training efficiency of caries image classification. By calculating mutual information between data samples and cluster centers, JSCDS effectively captures nonlinear dependencies in high-dimensional data, addressing the limitations of traditional methods. Experimental results demonstrate that JSCDS outperforms other data selection methods in prediction performance and time efficiency, achieving superior results with only 50-70% of the core data compared to the full dataset.
﻿
The paper presents the methodology clearly, and the experimental results are good.
﻿
Questions:
﻿
It is necessary to highlight the strengths and distinctive features of this method. What are the key advantages of this method compared to traditional and SOTA methods? Why does it outperform the methods cited [13][14][16], given that those methods can also model nonlinear relationships and high-dimensional data? Additionally, please conduct thorough research; the baseline methods from 2017, 2018, and 2020 need updating.
﻿
Why was JSD chosen? Is it a universal method for image analysis or specifically designed for this RGB dental caries dataset? If it is intended for general image data, more experiments are necessary. If designed for dental caries data, clarify the rationale for choosing this measurement method based on data characteristics.
﻿
Will reducing the amount of data affect the generalization of the deep learning models? Current evidence suggests that greater data quantity positively impacts model generalization and accuracy. In the experiments, the improved accuracy may be linked to dataset partitioning (as described in Section 3.1, with training, validation, and test sets in an 8:1:1 ratio). Consider incorporating cross-validation and external validation for further validation.

---

### Official Review · Reviewer_K8wF · 2024-06-21
**This paper makes incremental improvements on the Carie’s classification. The experimental results are preliminary.**

**Rating:** 6
**Confidence:** 3

**Review:**

This paper propose a Core Data Selection Method with Jensen-Shannon Divergence (JSCDS) for efficient caries image learning
and classification. More specifically, the proposed JDCDS selects the core data for model training by calculating the cluster centers of sample embedding representation. JSCDS captures nonlinear dependencies among high-dimensional data.

# Quality:
It is an acceptable paper, although the quality of the contents (e.g. typos, etc.) and figures should be improved (e.g. the font in the figure should be larger, etc.).

# Clarity:
The algorithm and its analyzing idea are missing in the paper, making readers not easy to grasp a good understanding to this work.

# Originality and Significance:
This is an incremental work on the caries classification.

# Strengths:
1.This paper proposes an information theory-based data selection based on average mutual information (AvgMI) to generate the core set, effectively distinguishing between different caries.
2.This paper proposes strategy that selecting core data by a mutual information calculation method with Jensen-Shannon Divergence which does not depend on model structure.
3. The proposed method exceeds the performance of the full dataset model with only 31 50% of the core data.

# Limitations:
1.The authors could discuss the differences between the proposed method and previous  caries classification methods based on deep learning architecture.
2.Some famous metrics for similarity, such as Manhattan Distance, Cosine Similarity, Minkowski distance, ect, should be employed for comparisons.
3.The ablation studies on JSD using different model structure should be verified.
4.The limitations and failure cases of the proposed method should be discussed.

---

### Decision · Program_Chairs · 2024-06-28

Accept (Poster)